# Investigation on Applying Biodegradable Material for Removal of Various Substances (Fluorides, Nitrates and Lead) from Water

**DOI:** 10.3390/ma16196519

**Published:** 2023-09-30

**Authors:** Ramunė Albrektienė-Plačakė, Kristina Bazienė, Justinas Gargasas

**Affiliations:** 1Department of Chemistry and Bioengineering, Vilnius Gediminas Technical University, 10223 Vilnius, Lithuania; ramune.albrektiene-placake@vilniustech.lt; 2Department of Mechanical and Material Engineering, Vilnius Gediminas Technical University, 10223 Vilnius, Lithuania; justinas.gargasas@vilniustech.lt

**Keywords:** sapropel, drinking water, fluoride, nitrate, lead, adsorption

## Abstract

Sapropel was used as a biodegradable material for water treatment. Sapropel is a sedimentary layer of a mix of organic and inorganic substances accumulated in the bottoms of lakes for thousands of years. It is a jelly-like homogeneous mass and has properties of sorption. Sapropel is used as a biosorbent and an environment-friendly fertiliser, and it is used in building materials and in the beauty industry as well. In water, there are abundant various solutes that may cause a risk to human health. Such substances include fluorides, nitrates and lead in different sources of water. The goal of this investigation is to explore and compare the efficiencies of removal of different pollutants (fluorides, nitrates and lead) from aqueous solutions upon using sapropel as a sorbent. In this research, various doses of sapropel (0.1, 0.5, 1, 5, 10, 20, 50, 100 and 200 g/L) and various mixing times (15, 30, 60, 90 and 120 min) were used for removal of fluorides, nitrates and lead from aqueous solutions. It was found that the maximum efficiency (up to 98.57%) of lead removal from aqueous solutions by sapropel was achieved when the minimum doses of it (0.1 and 0.5 g/L) were used. The most efficient removal of fluorides (64.67%) was achieved by using 200 g/L of sapropel and mixing for 120 min. However, sapropel does not adsorb nitrates from aqueous solutions.

## 1. Introduction

Sapropel is a sedimentary layer of a mix of organic and inorganic substances accumulated in the bottoms of lakes for thousands of years [1]. It is formed because of the decreasing level of groundwater, dying aquatic plants, leaves falling from onshore trees and the remains of plankton and benthos. This material is fine-grained and rich with organic matter sediment or sedimentary rock and refers to inland waters of the lacustrine environment [2]. Sapropel is a jelly-like homogeneous mass. The texture of the upper layers is cream-like, and the lower layers of sapropel become denser. The sediments are usually odourless, but sometimes the separate types of sapropel smell H_2_S. Organoleptically, sapropel can be from greenish-yellow (high silica content) to almost black (high organic matter and low mineral content), which depends on the site of exploration and the type of sapropel. Brown and dark green colour sapropel is found in Latvian lakes. This type of sapropel comes from the lake’s plants, plankton, and peat existence. The pH level of sapropel is from 7 to 8. These values of pH show that sediments have a high content of minerals [3]. It is formed in swallow-dying lakes and wetlands of lacustrine origin in their old-age period [4]. In Lithuania, its typical thickness is 1–2 m, sometimes up to 5 m. The stocks of pure sapropel are ~1 billion m^3^, and the stocks of this material mixed with impurities are ~6 billion m^3^ [5]. It is widespread throughout the world. The most intensive formation and accumulation of sapropel are typical for temperate climate zones in Asia and Europe (Poland, France, Germany, Russia, the Scandinavian Peninsula, the Baltic States, Ukraine and Belarus) as well as the Great Lakes Region in the North American continent (Canada and the United States) [6]. All the sapropels are divided into organic (50–90% of organic substances), calcareous (30–60% of calcium carbonate), silicon-based (25–45% of silica) and their mixes. There are several kinds (carbonated, peat, iron-rich, mixed, silicate with increased ashes contained, etc.) of each type. The main type is determined by the biological and oxide content [7]. Their chemical composition includes all the macroelements and microelements, as well as biologically active substances, such as ferments, vitamins and antibiotics, needed for plants [1,8,9]. The digging of sapropel, in addition to lake water treatment, enables the use of the collected sapropel as a biosorbent, an environment-friendly fertiliser, a feed additive, as well as in building materials and beauty industries [10,11,12]. The sorption process is an attractive wastewater treatment technology that does not require high costs or complicated sample preparation, especially if natural and environmentally rich materials are used [13]. One of the most important properties of wet sapropel is the suspended colloidal phase structure that enables organic colloidal particles of sapropel to absorb large quantities of water, so this material distinguishes itself for a high moisture capacity (70–97%) and a low filtering rate [12]. Because of the above-mentioned properties, sapropel may be used as a biosorbent. It was found that it can be used for the sorption of organic compounds and heavy metals [13,14]. In the course of research works, scientists found that the efficiency of lead removal while using sapropel was 81.6%, and the efficiency of sorption of zinc from a solution was higher—97.57% [13].

In various water sources, there are abundant different dissolved substances. Parts of them are useful; however, the other part of the said substances may pose a hazard to human health. In addition, substances related to pollution caused by a human or industry may appear in water as well. One type of substance most frequently appearing in groundwater in a natural way is presented by fluorides [15,16]. Fluoride ions can be found in minerals or water sources; Fluoride can occur in the food chain while drinking water or using vegetable food [17,18]. Fluorides can be useful or harmful to human health, which depends on their concentration in drinking water. They are useful for children to prevent tooth decay when the concentration of fluorides is 0.4–1 mg/L. However, an excessive amount of fluorides can lead to tooth or skeletal fluorosis [18,19,20,21]. Over 200 million people worldwide use drinking water with fluoride concentrations greater than 1.5 mg/L, the permissible value of the World Health Organization (WHO) for fluoride [22,23,24]. Fluorides can be removed from drinking water by the following methods: coagulation and precipitation methods [25], reverse osmosis and nanofiltration [26,27], ion-exchange method [28] and the adsorption technique [25,27,29,30,31].

In well-water, one type of the most frequent pollutants is represented by nitrates [32,33]. In a majority of locations worldwide, the nitrate concentration in water is increasing mostly because of the use of inorganic fertilisers and livestock manure in agriculture [34,35]. Nitrates are classified as toxic indicators, and a high concentration of nitrates may cause infant methemoglobinemia and cancers of the digestive tract in adults [36]. According to attitudes of the World Health Organization, the concentration of nitrates in drinking water should be less than 50 mg/L [37,38]. Nitrates can be removed from water by applying various removal technologies, namely: capacitive deionisation [39], ion exchange [40], by using electrochemical reduction with zero-valent titanium [41], electrodialysis [42], using biological nitrate removal [43] and by adsorption [44,45].

Lead is a toxic element that accumulates in the human body; it negatively affects the central nervous system and can cause dysfunction of the kidneys and the cardiovascular system [46,47,48,49]. This metal may be released into the environment from natural and anthropogenic sources. Natural pollution by lead appears because of volcanic eruptions and forest fires; however, industry and transport contribute to the pollution by lead as well [50,51,52]. It was found that in Poland, Japan, China, India, Singapore, Australia and Cambodia, the concentration of lead in drinking water was greater than the permissible values (0.01 mg/L, according to the World Health Organization) [49,53,54,55,56]. For lead removal from water, various technologies, such as adsorption, ion exchange, coagulation, precipitation and membrane technology [57,58,59,60], are used. In addition, lead can be removed upon applying various adsorbents, such as activated carbon, biomass, red clay, magnetite, chitosan, zeolites, aluminium, carbon nanotubes, bentonite, iron, manganese, zinc and copper oxides, aluminium sorbents, oak bark, mahogany bark, an amino bio-based resin derived from rosin, lignin, modified peat, rice shells, saw-dust, bamboo pulp, modified wool, active sewage sludge, palm tree waste fibres, nutshells, saffron flower waste, etc. [61,62,63,64,65,66,67,68,69,70].

Adsorption—the process of mass transfer by which atoms, ions or molecules are transferred from a liquid to the surface of a solid through chemical or physical interactions [71]. Adsorbate is the substance being adsorbed, and adsorbent is the adsorbing material. If the interaction between the surface of the solid and the adsorbed molecules is of a physical nature, the process is called physisorption [72]. The interactions between the van der Waals forces, however, are weak, and the processes are reversible. In the case of chemisorption, the attractive forces between the adsorbed molecules and the surface of the solid are due to ionic or covalent chemical bonds. In solid–liquid systems, adsorption consists of the removal of certain substances from the solution and their accumulation on the surface of the solid [73]. The amount of adsorbate that can be adsorbed depends on the temperature and the adsorbate concentration [72]. Physical adsorption is weakly specific, reversible and has a small thermal effect, while chemical adsorption is selective, generally irreversible and has a thermal effect ranging from tens to hundreds of kJ/mol [74,75]. During the adsorption process, the adsorbate migrates in three sequential steps: (1) migration of the adsorbate into the adsorbent boundary shell, (2) intraparticle diffusion into pores, and (3) adsorption and desorption of the solute [76]. The efficiency of the adsorption process for the removal of heavy metals is influenced by various factors, such as the initial concentration of metals, temperature, adsorbent dose, pH, contact time and stirring speed [77,78]. Adsorbents can be of mineral, organic or biological origin, e.g., activated carbon, zeolites, clay minerals, industrial by-products, agricultural waste, biomass and polymeric materials [79,80]. Conventional materials for adsorption of different pollutants are activated carbon, zeolites, clay minerals, industrial solid waste and biomaterials, such as chitin, clay, zeolite, peat moss, coal, and wood [81,82]. Modern sorbents, such as fullerenes [83], carbon nanotubes [84], graphene [85] and metal oxide-based nanomaterials [86], have unique physical, chemical and mechanical properties and reusability [87,88]. They are characterised by very high strength, toughness, electrical conductivity and thermal stability [87,88]. Adsorption of fluoride on a solid absorbent usually occurs in three phases: (1) diffusion or transfer of fluoride ions to the external surface of the adsorbent through a boundary layer surrounding the adsorbent particle, called external mass transfer; (2) adsorption of fluoride ions on the particle surface; (3) the adsorbed fluoride ions are probably exchanged with structural elements inside the adsorbent or the adsorbed fluoride ions move to the internal surfaces of the porous material [27,89]. The suitability of the adsorbent for the application depends on the adsorption capacity, selectivity for fluoride ions, regenerability, compatibility, particle and pore size, while the fluoride removal efficiency depends on the initial fluoride concentration, pH, temperature, contact time and the dose of adsorbent [88,89,90,91,92]. Metal oxides and hydroxides and their binary or tri-metal combination are effective in removing fluoride ions. The most commonly used are various oxides and hydroxides of titanium, iron and aluminium, which have the highest adsorptive properties over a wide pH range and high selectivity for fluoride ions. Biosorbents, such as chitin and chitosan, are the most commonly modified and adapted for fluoride removal. The adsorption efficiency of fluoride ions in biosorbents generally depends on the type of multifunctional group and the modification that has been made to increase the adsorption capacity [93].

A wide variety of materials have been used to absorb fluoride, nitrate and lead from aqueous solutions, and their mechanisms have been investigated, but very little research has been conducted on the ability of a natural biosorbent, sapropel, to absorb the different pollutants in water. The goal of this article is to explore and compare the efficiencies of removal of different pollutants (fluorides, nitrates and lead) from aqueous solutions upon using sapropel as a sorbent.

## 2. Materials and Methods

### 2.1. Materials

The raw sapropel for research was extracted from a 2–3 m depth of the Apslavas lake. The location of the lake is Stabulankliai village, Leliunai local municipality, Utena district, Lithuania. Figure 1 shows an image of dried sapropel.

The sample of sapropel was taken to determine its composition. The chemical (elemental) composition of sapropel was investigated using an X-ray fluorescence spectrometer with wave variance Axios Max (Almelo, Netherlands). An X-ray source of 4 kW power with rhodium anode was used. The chemical composition of sapropel was calculated using the Betalon methodology and Omniam software. The mineral composition of sapropel was investigated using powder X-ray diffraction diffractometer (SmartLab, Rigaku, Tokyo, Japan). Data analysis was made using the EVA software (Bruker AXS, Billerica, MA, USA) and the PDF-2 X-ray database. The carbon content of sapropel was determined by the CS-2000 carbon and sulphur analyser (ELTRA, Haan, Germany). 

The raw sapropel was washed twice with deionised water. It was dried for 3 h in a drying oven at 110 ± 2 °C after washing. The dried sapropel was crushed to a homogeneous mass and spread through a sieve (0.2 mm).

According to the provisions of WHO, the concentration of fluorides in water should not exceed 1.5 mg/L. The water samples were prepared at the laboratory using the standard fluoride solution c(NaF) = 1000 mg/L and distilled water. A total of 1 litre of the standard fluoride solution with a fluoride concentration of 3 mg/L was prepared. The said concentration was chosen on the basis of references [16].

According to the provisions of WHO, the concentration of nitrates in water should not exceed 50 mg/L. The water samples were prepared at the laboratory using sodium nitrate (NaNO_3_) and distilled water. A total of 1 litre of the solution with a nitrate concentration of 80 mg/L was prepared. The said concentration was chosen on the basis of references [33].

According to the provisions of WHO, the concentration of lead in water should not exceed 0.01 mg/L. The water samples were prepared at the laboratory using 1000 mg/L of the standard lead solution and distilled water. A total of 1 litre of the solution with a lead concentration of 0.07 mg/L was prepared. The said concentration was chosen on the basis of references [56].

### 2.2. Fluoride, Nitrate and Lead Removal Procedure

According to the scientific literature and practical knowledge, seven weighted amounts (1, 5, 10, 20, 50, 100 and 200 g) of sapropel were used for fluoride removal. Seven containers were taken, and 1 L of solution with a fluoride concentration of 3 mg/L in each container was added. Each weighted amount of sapropel was placed in those seven separate containers. Seven separate containers with test water were mixed with different doses of sapropel in a stirrer at 200 rpm. The water samples for fluoride determination were taken from each container in the quantities of 20 mL after 15, 30, 60, 90 and 120 min of mixing with different doses of sapropel. Each 20 mL of water sample after different duration mixing was filtered through a filter paper (47 mm diameter, the pore size of the membrane was 0.45 μm). After filtration of each 20 mL of water sample, tests of water quality for fluoride determination were performed.

For the removal of nitrates from the aqueous solution, six weighted amounts (1, 5, 10, 20, 100 and 200 g) of sapropel were used. Six containers were taken, and 1 L of solution with a fluoride concentration of 80 mg/L in each container was added. For the removal of lead from the aqueous solution, four weighted amounts (0.1, 0.5, 1 and 5 g) of sapropel were used. Four containers were taken, and 1 L of solution with a fluoride concentration of 0.07 mg/L in each container was added. The removal of nitrates and lead from aqueous solutions used exactly the same procedures and the same conditions as described for the removal of fluorides.

### 2.3. Analytical Methods

The concentration of fluoride in aqueous solutions was determined according to ISO 10359-1:1992, “Water quality—Determination of fluoride—Part 1: Electrochemical probe method for potable and lightly polluted water”.

The concentration of nitrate in aqueous solutions was determined according to ISO 7890-3:1988, “Water quality—Determination of nitrate—Part 3: Spectrometric method using sulfosalicylic acid“.

The concentration of lead in aqueous solutions was determined according to LST EN ISO 15586:2004, “Water quality—Determination of trace elements using atomic absorption spectrometry with graphite furnace“.

### 2.4. Statistical Methods

According to the “Council Directive 98/83/EC of 3 November 1998 on the quality of water intended for human consumption” [94], the result analysis is acceptable when the trueness and precision of the method do not exceed 10%. Trueness is a determination of systematic error, i.e., the difference between the mean value of the large number of repeated measurements and the true value. The stock solution of fluoride is 1.5 mg/L, of nitrate, it is 50 mg/L, and of lead, it is 10 μg/L according to the “Council Directive 98/83/EC of 3 November 1998 on the quality of water intended for human consumption”. The parametric value of fluoride is 1.5 mg/L, of nitrate it is 50 mg/L, and of lead, it is 10 μg/L [94]. Precision is a determination of random error and is usually expressed as the standard deviation of the spread of results from the mean. Acceptable precision is twice the relative standard deviation. The results of the analysis are expressed as the average concentration of 3 samples when the distribution is less than 10%. When the distribution is higher, the tests have to be repeated.

## 3. Results

The chemical (elemental) composition of sapropel was investigated using an X-ray fluorescence spectrometer with wave variance Axios Max. The mineral composition of sapropel was investigated using powder X-ray diffraction. The composition of sapropel is shown in Table 1.

Humidity—96.71% (drying 105 ± 2 °C), total nitrogen (N)—3.45% (Kjeldal method), sulphur (SO_3_)—0.9% (ICP-AS), organic matters—91.46% (burning to 500 °C), ashes—8.54% (calculated).

Figure 2 shows the microstructure of sapropel made by SEM. The sapropel particles are found to be irregularly shaped, unevenly distributed and containing debris.

In Figure 3, the efficiency of fluoride concentration decreases depending on the mixing time provided for two different doses of sapropel (1 and 5 g/L). The initial fluoride concentration is 3 mg/L. 

It can be seen from Figure 3 that when 1 and 5 g of sapropel are used, the changes in fluoride concentration are inconsiderable at different mixing times. If 1 g of sapropel is mixed for 15 min, the fluoride concentration falls from 3 to 2.8 mg/L (6.7%). At longer mixing times, the fluoride concentration gradually decreases; however, the decrease is inconsiderable. The best result is achieved when the mixing time is 120 min: in such a case, the fluoride concentration falls to 2.53 mg/L (15.7%). When a higher dose of sapropel (5 g/L) was used, no strong fluoride removal effect was observed. After 15 min, at different doses of sapropel, the fluoride concentration falls down to the same level—2.8 mg/L (6.7%). At mixing times of 30, 60 and 90 min, when a higher dose of sapropel (5 g/L) is used, the share of removed fluorides increases some more: from 2.8 to 8.6%. The best result is achieved when the mixing time is 120 min: in such a case, the fluoride concentration falls to 2.53 mg/L (15.7%) and is equal to the concentration when the used dose of sapropel is 1 g/L. When such doses of sapropel are used, the efficiency of fluoride removal at different mixing times is very low (up to 15.7% only), and the fluoride concentration is not less than the permissible norm of 1.5 mg/L.

Because the research works with low doses of sapropel (1 and 5 g/L) showed too low efficiency of fluoride removal, higher doses of sapropel (10, 20 and 50 g/L) were used in further works. In Figure 3, the dependence of the efficiency of fluoride concentration decreasing on mixing time when three different doses of sapropel (10, 20 and 50 g/L) are used is shown. The initial fluoride concentration is 3 mg/L.

It can be seen from Figure 4 that if the used dose of sapropel is 10 g/L and the mixing times are 15 and 30 min, the fluoride concentration slightly falls, as compared to the initial concentration—from 2.78 to 2.75 mg/L (7.33–8.33%). While increasing the contact time, the fluoride concentration gradually falls down from 2.44 to 2.17 mg/L (from 18.67 to 27.67%). However, if this dose is used, the fluoride concentration still is higher than permissible norms. If a higher dose of sapropel (20 g/L after 15 and 30 min mixing) is used, the efficiency of the fluoride concentration decreasing is higher as compared to the case when the dose of sapropel is 10 g/L, i.e., it falls down from 13.33 to 15.67%. On further mixing for 60, 90 and 120 min, the fluoride concentration remains practically the same as compared to cases where a lower dose of sapropel is used. The dose of 20 g/L is also not effective in fluoride removal at different mixing times. When the maximum dose of sapropel (50 g/L) is used, a higher efficiency of fluoride removal is observed already. On mixing for 15 and 30 min, the efficiency of fluoride removal achieves 33.33–36.33%, and the fluoride concentration goes down to 2 and 1.91 mg/L, respectively. At a longer mixing time (60 min), the efficiency of fluoride removal increases up to 48%, and the fluoride concentration falls to 1.56 mg/L. Such a concentration is very close already to the permissible norm—1.5 mg/L. While mixing for 90 and 120 min, the efficiency of fluoride removal increases up to 54–56.67%, and the fluoride concentration does not exceed the permissible norms anymore. The results show that when the used dose of sapropel is 50 g/L and the mixing time is 90 and 120 min, the most efficient fluoride removal is achieved, and the fluoride concentration does not exceed the permissible norms.

However, in the run of the research works, it was decided to further increase the doses of sapropel. In Figure 5, the efficiency of the decrease of fluoride concentration depending on the mixing time is provided when two different doses of sapropel (100 and 200 g/L) are used. The initial fluoride concentration is 3 mg/L. 

It can be seen from Figure 5 that if the used dose of sapropel is 100 g/L and the mixing time is 15 and 30 min, the fluoride concentration falls down to 2.2 mg/L (26.67%); however, it exceeds the permissible norms. On increasing the mixing time to 60, 90 and 120 min, the fluoride concentration gradually falls down from 1.45 to 1.16 mg/L (51.67–61.33%) and does not exceed the permissible norms anymore. When the maximum concentration of sapropel (200 g/L) is used and the mixing time is 15 and 30 min, the fluoride concentration falls down to 1.5 mg/L (50%) and is close to the detection limit. If the contact time is longer (60, 90 and 120 min), the fluoride concentration gradually falls down from 1.39 to 1.06 mg/L (53.67–64.67%) and does not exceed the permissible norms anymore.

After completion of the tests upon applying different doses of sapropel and different mixing times, it was found that the fluoride concentration (1.38 mg/L) did not exceed the permissible norms when the used dose of sapropel was 50 g/L and the contact time was 90 min. The best efficiency of fluoride removal (64.67%) is achieved with 200 mg/L of sapropel and with the 120 min contact time.

In addition, similar tests were carried out upon trying to remove nitrates from water. In the first stage, the minimum doses of sapropel (1 and 5 g/L) were chosen. In Figure 6, the efficiency of the decrease of nitrate concentration depending on the mixing time is provided when two different doses of sapropel (1 and 5 g/L) are used. The initial nitrate concentration is 80 mg/L. 

It can be seen from Figure 6 that when the doses of sapropel (1 and 5 g/L) are used, the nitrate concentration does not fall at mixing for 15 min or even for 120 min. Therefore, it was decided to increase the doses of sapropel up to 10 and 20 g/L. In Figure 7, the efficiency of the decrease of nitrate concentration depending on the mixing time is provided when two different doses of sapropel (10 and 20 g/L) are used. The initial nitrate concentration is 80 mg/L.

While testing the sorption when higher doses of sapropel (10 and 20 g/L) are used, the results of the test show that the testing time provides no impact on nitrate removal. If sapropel is mixed with water contaminated by nitrates, changes in the nitrate concentration appear neither after mixing for 15 min nor after 120 min. Therefore, it was decided to increase the doses of sapropel up to 100 and 200 g/L.

In Figure 8, the efficiency of the decrease of nitrate concentration depending on the mixing time is provided when two different doses of sapropel (100 and 200 g/L) are used. The initial nitrate concentration is 80 mg/L.

After completion of the tests of the sorption time when higher doses of sapropel (100 and 200 g/L) were used, the results of the tests show that sapropel provides no impact on nitrate removal. No sorption takes place. If sapropel is mixed with water contaminated by nitrates, changes in the nitrate concentration appear neither after mixing for 15 min nor after 120 min.

In addition, similar tests were carried out upon trying to remove lead from water. In the first stage, the minimum doses of sapropel (1 and 5 g/L) were chosen. In Figure 9, the efficiency of the decrease of lead concentration depending on the mixing time is provided when two different doses of sapropel (1 and 5 g/L) are used. The initial lead concentration is 70 μg/L. 

It can be seen from the results of the tests shown in Figure 9 above that the minimum doses of sapropel at the shortest mixing times show very high efficiency of lead removal from water. If a sapropel dose of 1 g/L is used, the lead concentration after 15 and 30 min falls down to 4–5 μg/L, and the efficiency of lead removal achieves 92.86–94.29%. If the sorption time is increased to 60–120 min, the lead concentration practically decreases by 100%. If a higher dose of sapropel (5 g/L) is used, the lead concentration decreases by 100% at different sorption times. Although such a high efficiency of lead removal from water was found, it was decided to reduce doses of sapropel.

In Figure 10, the efficiency of the decrease of lead concentration depending on the mixing time is provided when two different doses of sapropel (0.1 and 0.5 g/L) are used. The initial lead concentration is 70 μg/L.

We can see from Figure 10 above that when the doses of sapropel are reduced to 0.1 and 0.5 g/L, the efficiency of the reduction of lead concentration is very high. After 15 min of mixing, if two said doses are used, the lead concentration falls down to the same level of 2 μg/L and a 97.14% efficiency of removal is achieved. At a longer mixing time, the lead concentration fluctuates within the error margins from 1 to 4 μg/L. The efficiency of lead removal from 94.29% to 57% was achieved. If all doses of sapropel (even the minimum ones) and different mixing times are used, the lead concentration does not exceed the permissible limit of 10 μg/L. So, it may be stated that 0.1 g/L dose of sapropel and 15 min mixing time are sufficient for lead removal from water to ensure the permissible norm.

## 4. Discussion

In the research, cheap biosorbent sapropel was used. It was used for the removal of three different contaminants (fluorides, nitrates and lead) that cause a strong negative influence on the environment and human health from aqueous solutions. The results of the research showed very different efficiencies of removal of the said contaminants from aqueous solutions. Sapropel practically did not adsorb nitrates; however, in the case of lead, the efficiency of its removal achieved 98.57% (similar high efficiencies of lead removal obtained by other researchers: 81.6% [13] and 98% [14]). How can such a difference be explained? First of all, it is necessary to clear up the differences between fluorides, nitrates and lead and what substances the sapropel consists of. Fluorides and nitrates in an aqueous solution are presented as negative ions (F^−^ and NO_3_^−^), and lead in an aqueous solution is usually a positive ion Pb^2+^. Anions and cations are removed from aqueous solutions in different ways. 

Scientists found that fumed oxides (silica and mixed oxides based on silica) are efficient adsorbents for toxic metals, such as Ni, Pb, Cd, Sr, and Cs [95]. The use of inorganic materials, especially silica, has a high surface area to enhance the capacity of adsorption and great chemical and physical robustness to withstand a diversity of harsh environments. The excellent adsorbent for the adsorption of Cu and Pb is highly structured mesoporous silica with incorporated bridging/complex-forming functional mercapto or amino groups [96]. As early as 1960, it was considered, starting with the scientist Pauling, that there might be a pπ–dπ interaction between silicon atoms and elements which have free electron pairs. It has been identified that there is an extra π-interaction in siloxane bonds SiO, which is the cause of significant covalence of this bond [97,98,99,100]. The degree of this covalent bond is determined by the nature of the cation bound to the oxygen atom. This shows that the bond O–Me^n+^ in functional group Si–O–Me^n+^ should have some covalence, the degree of which increases with an increase in the acceptor ability of the metal ion [100]. This happens due to competition between the ions of metal and silicon for the free electron pair on the oxygen atom. According to data provided in the literature, the principal mechanism of cation adsorption in silica gel is ion exchange with surface hydrogen ions of silanol groups, in particular at low pH values. Adsorption of heavy and transition metal ions surfaces functionalised with silica gel is provoked by the interaction of different characters (hydrogen bonds, electrostatic, complex-forming, etc.) [101]. As we can see from scientific sources, silica is distinguished for good adsorbing properties in the removal of ions of toxic metals from aqueous solutions. While looking at the chemical composition of sapropel provided in Table 1, we can see that 65% of it consists of a SiO_2_ compound. Therefore, we can state that such good results of lead adsorption efficiency are achieved because of SiO_2_ (which is distinguished for good sorption properties with respect to heavy metals).

In order to clear up why sapropel removes fluorides and nitrates to a lesser extent or does not remove them at all, we would find out what compounds remove anions from aqueous solutions best. According to scientists, Al-OH and Fe-OH groups are very important for anion adsorption. Because of a large surface area, the efficiency of anion adsorption by hydrous iron and aluminium oxides is very high. Anion adsorption involves an electrostatic interaction as well as some chemical interaction between the surface and the ion. Anion adsorption depends on the pH value; the maximum adsorption takes place when fully dissociated ions are formed at low pH values, and the surface becomes positively charged because of the protonation of Me-OH groups on it. Anion adsorption on the said surfaces occurs as follows: phosphate > arsenate > selenite = molybdate > sulphate = fluoride > chloride > nitrate. More adsorbed anions react with Me-OH_2_^+^ and Me-OH groups in a ligand exchange reaction where anion becomes coordinated with metal ions [102]. The soils with a top content of iron are goethite (α-FeOOH) and hematite (α-Fe_2_O_3_). We can see in the chemical composition of sapropel provided in Table 1 that Fe_2_O_3_ (type of hematite) predominates in sapropel (0.94%), and it probably influences anion adsorption. Upon taking into account the sequence of anion adsorption described by scientists (phosphate > arsenate > selenite = molybdate > sulphate = fluoride > chloride > nitrate) [102], we can see that fluorides on surfaces with Fe-OH groups are adsorbed better than nitrates. On the said surfaces, nitrates are adsorbed at the end of the adsorption sequence. Therefore, we can see that sapropel adsorbs fluoride anions better than nitrate anions. The low efficiency of removal of fluoride anions can be explained as follows: the content of iron compounds that adsorb anions well is low in sapropel; therefore, large doses of sapropel are required for achieving the necessary removal of fluorides.

## 5. Conclusions

When sapropel was used as a cheap bioadsorbent for the removal of fluorides, nitrates and lead from aqueous solutions, it was found that in case of removal of fluorides from aqueous solutions (if the initial concentration is 3 mg/L) using different doses of sapropel and different mixing times, the fluoride concentration falls to the required permissible norms for drinking water when the used dose of sapropel is 50 g/L and the mixing time is 90 min. The biggest efficiency of fluoride removal (64.67%) is achieved by using 200 g/L of sapropel and mixing 120 min. Sapropel is not fit for the removal of nitrates from aqueous solutions because it does not reduce the nitrate concentration in an aqueous solution. When minimum doses of sapropel (0.1 and 0.5 g/L) are used, the efficiency of lead removal is up to 98.57%. Using any (even the smallest) doses of sapropel and different mixing times, the lead concentration does not exceed the permissible level of 10 μg/L set for drinking water. Such different efficiencies of removal of the said contaminants from water are caused by the circumstance that anions and cations are removed from the water, and their sorption mechanisms differ; in addition, the composition of sapropel varies as well.

Our study shows that sapropel is actually very effective at removing lead from water. It also removes fluoride but requires higher doses and longer mixing times. And it does not remove nitrates at all. Therefore, sapropel can be further investigated as a potential sorbent of lead and fluoride from aqueous solutions. Further studies should investigate which conditions are the most favourable for the removal of lead and fluoride, how long the sapropel can absorb these compounds and how and at what frequency it should be regenerated. In addition, the feasibility of using sapropel in prototype water filters with different technical modifications could be investigated under laboratory conditions.

## Figures and Tables

**Figure 1 materials-16-06519-f001:**
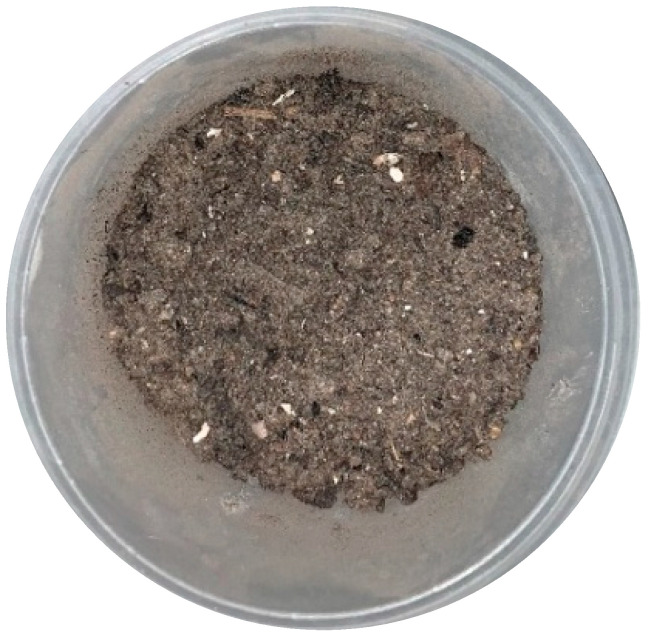
The image of dried sapropel.

**Figure 2 materials-16-06519-f002:**
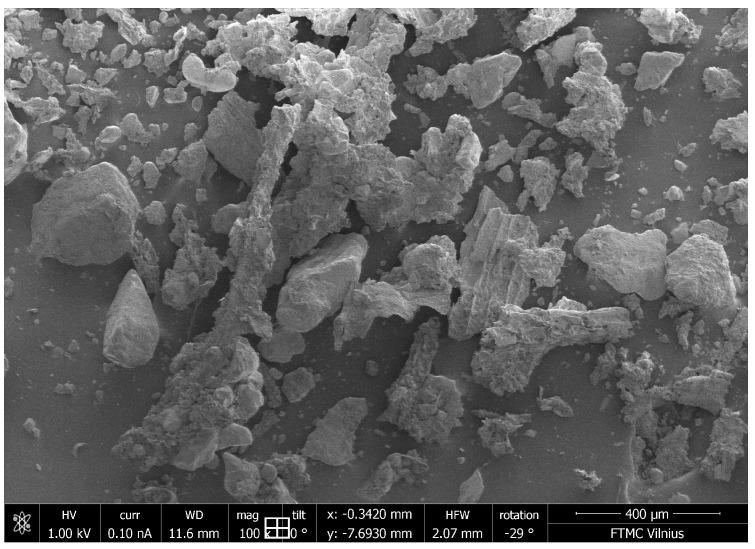
Microstructure of sapropel.

**Figure 3 materials-16-06519-f003:**
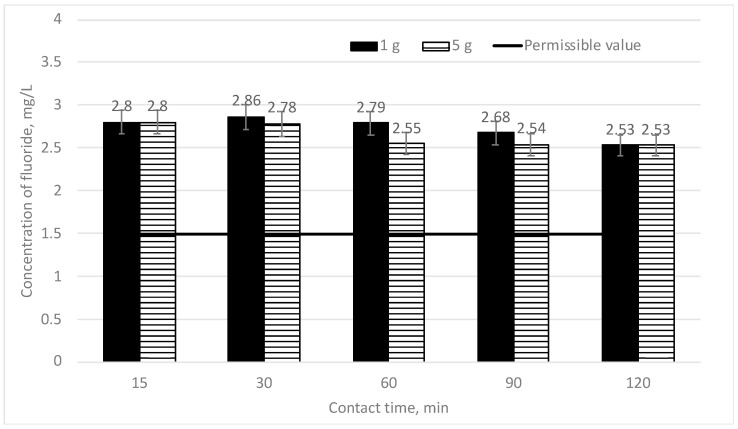
The dependence of fluoride concentration on sorption contact time when the doses of sapropel are 1 and 5 g/L.

**Figure 4 materials-16-06519-f004:**
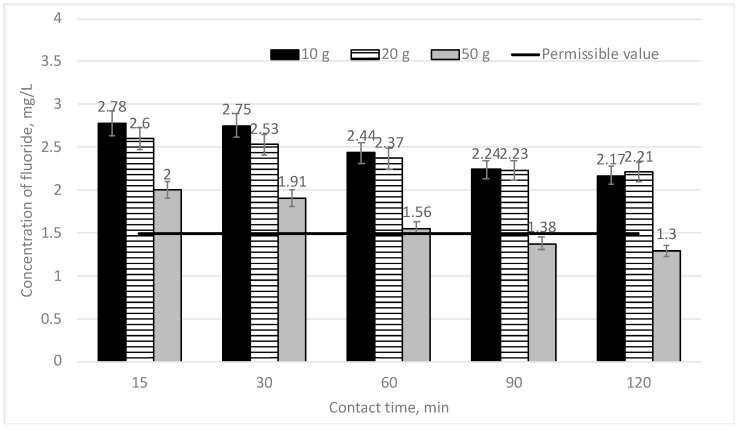
The dependence of the fluoride concentration on sorption contact time when doses of sapropel are 10, 20 and 50 g/L.

**Figure 5 materials-16-06519-f005:**
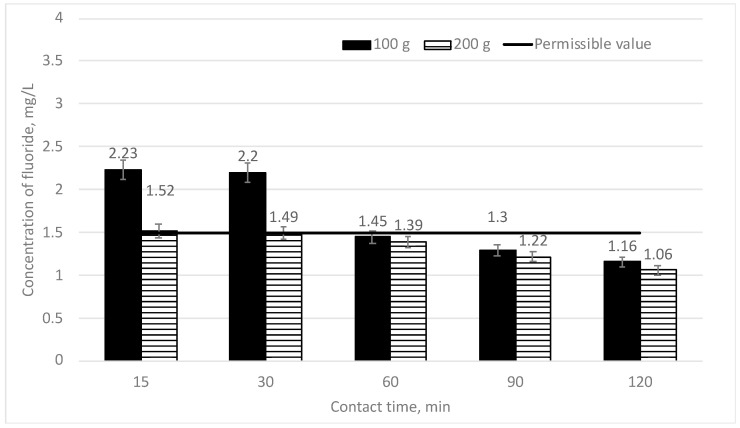
The dependence of fluoride concentration on sorption contact time when doses of sapropel are 100 and 200 g/L.

**Figure 6 materials-16-06519-f006:**
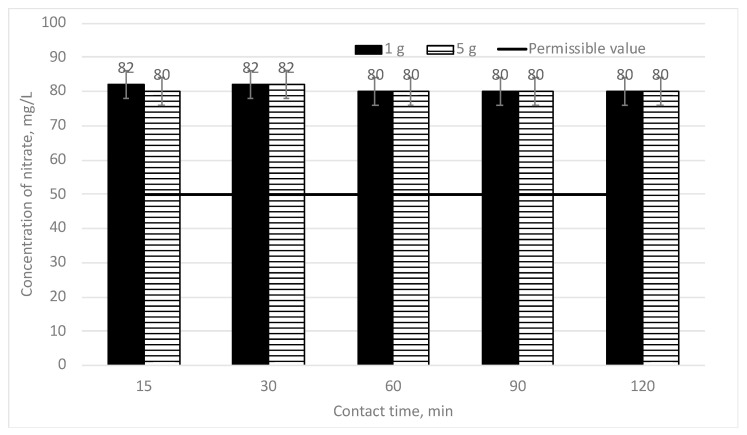
The dependence of nitrate concentration on sorption contact time when the doses of sapropel are 1 and 5 g/L.

**Figure 7 materials-16-06519-f007:**
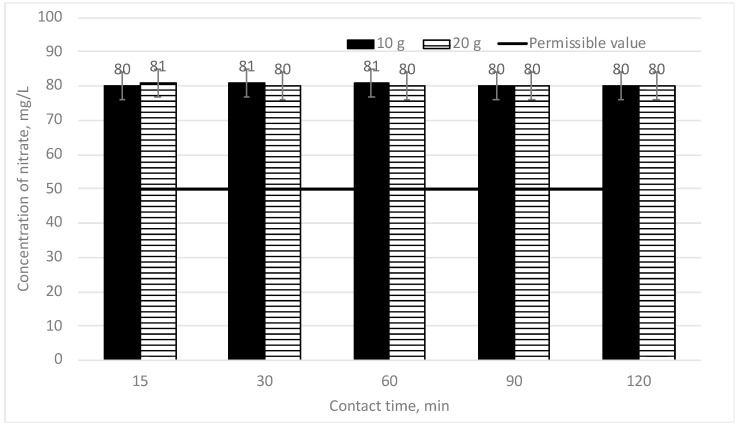
The dependence of nitrate concentration on sorption contact time when the doses of sapropel are 10 and 20 g/L.

**Figure 8 materials-16-06519-f008:**
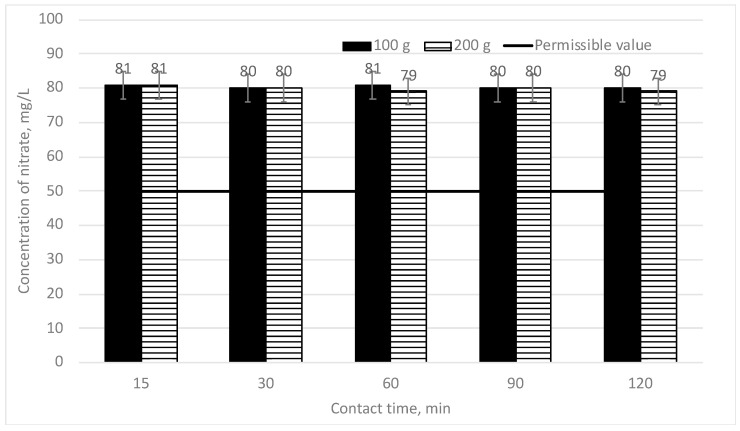
The dependence of nitrate concentration on sorption contact time when the doses of sapropel are 100 and 200 g/L.

**Figure 9 materials-16-06519-f009:**
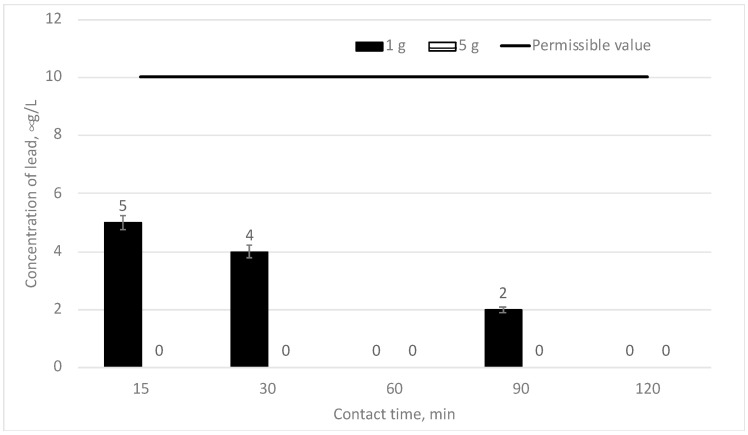
The dependence of lead concentration on sorption contact time when the doses of sapropel are 1 and 5 g/L.

**Figure 10 materials-16-06519-f010:**
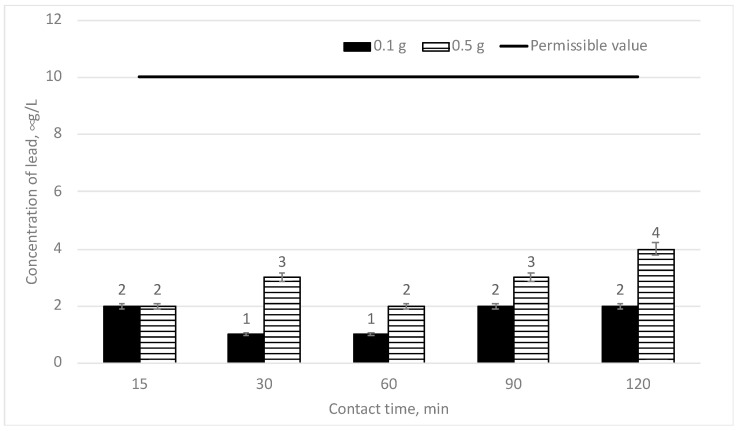
The dependence of lead concentration on sorption contact time when the doses of sapropel are 0.1 and 0.5 g/L.

**Table 1 materials-16-06519-t001:** The chemical composition data of sapropel.

Element	*w*/*w*,%	Compound	*w*/*w*,%
C	6.616	CO_2_	24.240
O	56.325	O	0.105
Si	30.383	SiO_2_	65.000
Mg	0.279	MgO	0.462
Na	0.363	Na_2_O	0.489
Al	2.602	Al_2_O_3_	4.916
S	0.165	SO_3_	0.413
P	0.040	P_2_O_5_	0.091
Ca	0.890	CaO	1.246
K	1.422	K_2_O	1.712
Cl	0.011	Cl	0.011
Ti	0.147	TiO_2_	0.245
Cr	0.027	Cr_2_O_3_	0.040
Fe	0.657	Fe_2_O_3_	0.940
Mn	0.010	MnO	0.013
Cu	0.001	CuO	0.001
Ni	0.003	NiO	0.003
Zn	0.003	ZnO	0.003
Pb	0.007	PbO	0.008
Zr	0.024	ZrO_2_	0.032
Sr	0.005	SrO	0.006
Rb	0.005	Rb_2_O	0.006
Ba	0.013	BaO	0.015
Y	0.002	Y_2_O_3_	0.002

## Data Availability

Not applicable.

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
