# Peer review of "Investigation on Applying Biodegradable Material for Removal of Various Substances (Fluorides, Nitrates and Lead) from Water"

_materials, 2023, doi:10.3390/ma16196519_

Round 1

Reviewer 1 Report

The research seems interesting but does not look impactful with the level of work presented. It would be good idea to see how it behaves in the the solution with mixture of contaminants rather than just one individual contaminant. Ideally no water will have only one single contaminant to be removed. So to justify the validity of work for the real world perspective, more work need to done in this research presented and may be see how it behaves in some real water samples. If authors could include some additional work regarding how impactful is it for real application that would be valuable.

Author Response

Response to Reviewer 1 Comments

We are very grateful for your opinion and valuable remarks. The manuscript has been revised taking into

account all these comments. 

The research seems interesting but does not look impactful with the level of work presented. It would be good idea to see how it behaves in the the solution with mixture of contaminants rather than just one individual contaminant. Ideally no water will have only one single contaminant to be removed. So to justify the validity of work for the real world perspective, more work need to done in this research presented and may be see how it behaves in some real water samples. If authors could include some additional work regarding how impactful is it for real application that would be valuable.

Authors comment: Thank you for your opinion and suggestion. We have thought about that as well. But all components (fluoride, lead and nitrate) appear in very different sources of water. Fluoride appears mostly in groundwater, lead  - in surface water and nitrate – in well-water. All types of water’s sources are very different compositions of other components, which can influence the absorption of our components. Even if we would choose, for example groundwater, different deep wells of groundwater are totally different from each other compared to chemical composition. We should take into account more than 50 components (according Council Directive 98/83/EC). If we would present an analysis of one real water sample with its characteristic chemical composition, it didn’t fit for other water samples. It would be researched only for one source of water. We wanted to find out if sapropel is useful for removing some different components from water in general.

Reviewer 2 Report

Journal: Materials

Ms. ID.: materials-2615846

Title: Water treatment from fluorides, nitrates and lead using biodegradable material

Albrektienė-Plačakė et al aim to to investigate the effectiveness of sapropel, a biodegradable material derived from lake sediments, as a biosorbent for the removal of specific water contaminants, namely fluorides, nitrates, and lead, which can pose risks to human health. The study explores various doses of sapropel and mixing times to determine the optimal conditions for the removal of these contaminants from aqueous solutions. The research seeks to provide insights into the potential use of sapropel as an eco-friendly and efficient method for water treatment and contamination mitigation. The subject is interesting and fits well with the scope of the journal. Still, the manuscript is poorly prepared. Here are my comments:

-The title is not suitable since the material does not adsorb nitrates. It should be revised.

-While the abstract provides an overview of the study, it could benefit from greater clarity and conciseness. Some sentences are relatively long and could be broken down into shorter, more easily digestible sentences. It would be helpful to provide a brief explanation or context for sapropel in the abstract itself, as not all readers may be familiar with this material. This would ensure a better understanding of the biosorbent being used. Also, the abstract lacks a clear statement of the research objectives or hypotheses. Including a sentence that explicitly states the aim of the study would provide better context.

-The introduction is not well-balanced and needs revision. It should be less about the sapropel and more about the adsorption process for water purification. The adsorption is not discussed adequately. Also, the language needs to be revised since the word sapropel is mentioned many times.

-Characterization of sapropel needs to be moved in the Results section.

-The results are presented very monotonously. The discussion on the mechanism of the adsorption is merely a guess. The authors should try FTIR analysis before and after the adsorption in order to obtain more information.

-The conclusion mentions the efficiency of sapropel in removing contaminants but does not delve into the practical implications of these findings. It is important to discuss how these results could be applied or what they mean in a broader context, such as in water treatment processes or environmental remediation. It's helpful to reiterate in the conclusion how the study's findings relate to the initial research objectives or hypotheses. This can provide a sense of closure and alignment with the study's purpose. The authors should include a brief mention of potential avenues for future research or further investigation based on the current findings. 

Moderate editing is needed.

Author Response

We are very grateful for your opinion and valuable remarks. The manuscript has been revised taking into account all these comments. 

Round 2

Reviewer 2 Report

The authors addressed most of my comments, but the Introduction still needs to be revised. The process of adsorption as a water purification technique in general, needs to be discussed. 

Moderate changes are required. 

Author Response

Thank you for your comments. The process of adsorption was more detail discussed.
